# The Yeast Atlas of Appalachia: Species and Phenotypic Diversity of Herbicide Resistance in Wild Yeast

**Jordan B. Barney, Matthew J. Winans, Catherine B. Blackwood †, Amaury Pupo and Jennifer E.G. Gallagher \***

Department of Biology, West Virginia University, 53 Campus Drive, Life Sciences Building, Morgantown, WV 26506, USA; jbbarney@mix.wvu.edu (J.B.B.); mwinans@mix.wvu.edu (M.J.W.); cblackwo@mix.wvu.edu (C.B.B.); amaury.pupo@gmail.com (A.P.)

**\*** Correspondence: jegallagher@mail.wvu.edu; Tel.: +1-304-293-5114; Fax: +1-304-293-6363

**†** Current Address: Department of Microbiology, West Virginia University, Immunology and Cell Biology, 1 Medical Center Drive Morgantown, WV 26506-9177, USA.

**Abstract:** Glyphosate and copper-based herbicides/fungicides affect non-target organisms, and these incidental exposures can impact microbial populations. In this study, glyphosate resistance was found in the historical collection of *S. cerevisiae*, which was collected over the last century, but only in yeast isolated after the introduction of glyphosate. Although herbicide application was not recorded, the highest glyphosate-resistant *S. cerevisiae* were isolated from agricultural sites. In an effort to assess glyphosate resistance and impact on non-target microorganisms, different yeast species were harvested from 15 areas with known herbicidal histories, including an organic farm, conventional farm, remediated coal mine, suburban locations, state park, and a national forest. Yeast representing 23 genera were isolated from 237 samples of plant, soil, spontaneous fermentation, nut, flower, fruit, feces, and tree material samples. *Saccharomyces*, *Candida*, *Metschnikowia*, *Kluyveromyces*, *Hanseniaspora*, and *Pichia* were other genera commonly found across our sampled environments. Managed areas had less species diversity, and at the brewery only *Saccharomyces* and *Pichia* were isolated. A conventional farm growing RoundUp Ready™ corn had the lowest phylogenetic diversity and the highest glyphosate resistance. The mine was sprayed with multiple herbicides including a commercial formulation of glyphosate; however, the *S. cerevisiae* did not have elevated glyphosate resistance. In contrast to the conventional farm, the mine was exposed to glyphosate only one year prior to sample isolation. Glyphosate resistance is an example of the anthropogenic selection of nontarget organisms.

**Keywords:** fungal diversity; *Saccharomyces*; genetic diversity; glyphosate-based herbicides (GBH); copper-based fungicides; RoundUp Ready™ corn; phylogenetics

## 1. Introduction

### 1.1. Diversity of Yeast

While yeasts have become a commonplace model organism in laboratory experiments, research on the ecology, natural habitats, and genetic diversity of yeasts is relatively recent. As a model for evolutionary genetics studies, the *Saccharomyces* clade is one of the most studied clades of yeast, but the diversity within the genus has a complex history as the species in this clade readily hybridize with other species and are morphologically similar [1–4]. Determining species by morphology or mating/sporulation is laborious and inherently prone to mistaken classifications, resulting in strains

being classified as species. Collections of yeasts isolated from a variety of substrates all over the world are used in laboratory research [2,5,6]. While this collection of yeast provides a wide array of genetically diverse strains for examination and experimentation, the library itself typically has one sequenced representative strain from each location and does not address the hidden yeast diversity [7]. Increasing sampling depth at a site would address questions on a finer ecological scale, such as species richness and within-species diversity. We measured the phylogenetic diversity of different species of yeasts in environments such as forests, compared to the diversity of yeasts in metropolitan areas [8]. *S. cerevisiae*, a member of the sensu stricto clade (defined as yeast that can form hybrids with *S. cerevisiae*), was the first eukaryotic organism to have its genome entirely sequenced [9]. At least 30% of its genes have homologs within the human genome and to varying degrees, the human homologs can function in yeast [10]. For this reason, yeasts are considered a model organism for a variety of experiments. For example, incorporation of *S. kudriavzevii* into this research has offered insights on how yeast evolve to survive in new environments and also hybridization leads to the formation of new species [11]. Hybridization of *S. cerevisiae* with *S. arboricola* expands the flavor and increases alcohol production of sake [12]. The development of collections that include species beside *Saccharomyces cerevisiae* will help to expand insights into yeast evolution and the ability of yeast to withstand different environments [13]. Because yeasts primarily metabolize sugars, they are readily available in sources such as flowers, fruits, leaves, and bark extrudates. While the relationships between *S. cerevisiae* and its hosts are not thoroughly defined, there is evidence that shows an association between *Saccharomyces* clade species and the *Quercus* genus of trees [14,15]. Additionally, the wintering process among species of *Saccharomyces* has been shown to occur within wasps, beetles, and other insects that facilitate survival in the insect gut during winter and their redistribution during warm spring temperatures [15–17]. The fermentation of alcoholic beverages is dominated by *S. cerevisiae* and interspecies hybrids. This fermentation industry has domesticated yeast specifically for certain brews and *S. cerevisiae* is seen throughout the brewing process for a variety of fermented beverages [7,18].

*1.2. Glyphosate-Based Herbicides*

Glyphosate is the active ingredient in the herbicide RoundUp™, which has been used globally since its introduction in the 1970s. Including RoundUp, the use of herbicides has predictable consequences, such as an eventual adaption to herbicide resistance in plants that have been previously exposed [19–22]. While many crop plants have been genetically modified to have resistance to the herbicide [23–25], the genetic changes that have occurred in weeds to propagate this resistance have been challenging to determine. Glyphosate prevents the synthesis of aromatic amino acids by inhibiting the shikimate biosynthetic pathway [22]. It affects nearly all plants, yeasts, and bacteria that synthesize aromatic amino acids, para-aminobenzoic acid, and co-enzyme Q using the shikimate pathway. In yeast, the variation of ABC drug transporter Pdr5 and the glutamic/aspartic acid permease Dip5 change their tolerance to glyphosate-based herbicides (GBH) [26]. However, in the over 300 commercial formulations, glyphosate is not the only ingredient. Additives improve the action of glyphosate and have effects on cellular metabolism on their own [27,28]. By changing the growth media, the effect of the additives can be separated. In yeast, Aro1 is the ortholog of EPSPS, the target protein of glyphosate. Aro1 catalyzes the rate-limiting step in the synthesis of chorismate, the precursor for all aromatic compounds—including tryptophan, tyrosine, and phenylalanine (WYF). Inhibition of growth in minimal media by glyphosate is due to inhibition of Aro1 in the canonical pathway and this can be bypassed by supplementation with WYF [26]. In contrast, inhibition of growth in WYF with glyphosate points to inhibition of a non-canonical target. Between genetically diverse yeasts, there are tens of thousands of SNPs [29]. Quantitative trait loci analysis of glyphosate resistance shows that it is a polygenetic trait [26–28] and there are multiple changes that can occur to select for glyphosate resistance [27,28]. Through QTL and in-lab-evolution studies, the additives in glyphosate based herbicides (GBH) affect diverse pathways such as cell wall and mitochondria [27]. Commercial preparations also contain detergents as additives that have additional effects such as cell wall stress

and cell cycle arrest [27]. Rich media, such as YPD, is an undefined media that contains all amino acids, fatty acids, and other biomolecules. *S. cerevisiae* that are sensitive to the commercial preparations in rich media are due to the detergents rather than from glyphosate [27].

*1.3. Copper-Based Herbicides*

Metal ions such as copper have been used as fungicides and antimicrobials for centuries and continue to be employed in organic farming [30]. This toxicity can be beneficially exploited in applications involving the prevention of bacterial infection in medical equipment. Copper sulfate (CuSO4) has been applied to fight fungal blights in berries, pomes, stone fruits, and walnuts. Many chemicals derived from natural sources are utilized for organic farming; copper is one example [31]. Widespread use of copper prevents microbial growth because of its broad-spectrum activity and multiple modes of toxicity [32]. Excessive copper likely induces damage by catalyzing reactive oxygen species (ROS) generation [33]. ROS generated by extracellular copper directly oxidizes lipids and proteins of the cell membrane [34,35], and if cells survive this damage then DNA damage can occur [36]. The multiple modes of toxicity and the response to copper in microorganisms has been widely studied [37–39]. Excessive intracellular copper is absorbed by the copper metalloproteins Cup1 and Crs5 [40–42]. Ctr2 is a copper transporter that brings excess copper to the vacuole [43]. In copper excess, the transcription factors, Ace1 and Ace2, induce transcription of *CUP1*, *CRS5*, and *SOD1* [44–47]. The *CUP1* locus is highly variable across strains with local amplifications and the existence of paralogs [42,48]. The expression of genes in response to copper is highly regulated to maintain a fine balance of this essential-yet-toxic trace mineral and it uses very different mechanisms than GBH resistance.

Within *S. cerevisiae*, there is a wide range of tolerance to GBH and genetic analysis between three difference strains have found variation in amino acid permeases, pleiotropic drug transporters, cell wall proteins, and proteins relating to mitochondrial function [26–28]. One of these previously characterized strains is an agricultural strain while the others are a common lab strain and a clinical isolate. We surveyed a historical collection of *S. cerevisiae* to determine if agricultural isolates were more likely to be resistant to glyphosate-based herbicides. We expand this survey to *S. cerevisiae* isolated from areas with known GBH exposure. In the process, we also cataloged the fungal diversity in the wild by sequencing the rDNA loci to determine genus and species, and then assessed resistance to glyphosate of *S. cerevisiae* isolates. We also measured copper resistance of the same isolates because none of the collection sites were known to be exposed intentionally to copper. Copper and glyphosate resistance are fundamentally different which would serve as a comparison to GBH resistance. The locations were divided into a 'managed' or 'pristine' categories. Managed areas have some human management, such as the application of glyphosate-based herbicides. Pristine areas have never been exposed to herbicides, to the best of our knowledge, or are so remote that it is unlikely that herbicides have ever been used in these areas. We found that more yeast species were isolated from pristine areas, while managed areas were dominated by fewer species. Flowers yielded the most yeast species diversity. *S. cerevisiae* itself was most often isolated from trees, followed by soil, then flowers. Of the groups of substrates identified, animal and air had nearly a perfect isolation success of yeasts but were of a small sample size. From nearly 100 samples, tree substrates produced a yeast isolate in three of every four. Yeast isolated from RoundUp Ready™ corn were the most glyphosate resistant. None of the areas were known to be exposed to copper and a few *S. cerevisiae* strains displayed tolerance to high levels of copper. Agricultural yeast from the historical collection isolated after the 1980s were associated with high glyphosate tolerance. The historical collection was curated from previous publications to encompass geographic and niche diversity of sequenced yeast [5]. The remediated strip mine was heavily sprayed with herbicides starting the year before collections began and in contrast to the conventional farm, the mine had minimal glyphosate exposure with less glyphosate resistance. In addition, there was more genetic diversity from yeast collected from the forest surrounding the mine than from the mine itself. The phenotypic profile of the yeast was able to discriminate between *S. cerevisiae* isolated from

the mine from the forest. Consistent with previously isolated yeast, agricultural isolates that have years of glyphosate exposure display the highest tolerance. This demonstrates the effect of manmade selective pressures on wild yeast.

## 2. Materials and Methods

### 2.1. Collection of Samples

Sterile polyethylene bags were used to collect plant material from bark, extrudes, fruits, flowers, animal feces, and soils beneath previously fruiting plants or trees. Samples were collected from the following areas within West Virginia: Appalachian Trail, Chestnut Brew Works, Coopers Rock, WVU Campus, Fernow Forest, Jefferson County, Osage Remediated Coal Mine, a commercial farm, Granville, Spruce Knob Mountain, WVU Organic Farm, WV Botanical Garden, and the WVU Arboretum, as well as the Ohiopyle State Park in Pennsylvania, and a home garden in Jalisco, Mexico. GPS coordinates of each sample collection points were recorded using the mobile device of the individual collecting the sample and through the use of several mobile phone applications (Supplemental Table S1). Coordinates were recorded and later used to create the map of collection sites. Vector maps were created with R packages albersusa 0.3.1, ggplot2 3.2.1, and ggrepel 0.8.1. The simple feature standard objects corresponding to the states of Pennsylvania, West Virginia, Virginia, and Maryland were taken from albersusa and merged. The maps and locations were plotted with ggplot2 and the labels added with ggrepel.

### 2.2. Isolation of Yeast-Like Colonies

To extract yeast from the material collected, each sample was soaked in enrichment media containing 3 g yeast extract, 3 g malt extract, 5 g peptone, 10 g sucrose, 76 mL ethanol, and 1 mL HCl per liter for 10 min in the bag in which it was collected [14]. After incubation, roughly 2 mL was transferred to a 15 mL tube and treated with 10 mg/mL kanamycin, 1 mg/mL chloramphenicol, and 1 mg/mL ampicillin, before being incubated at 24 °C to 30 °C for 7–10 days, or until signs of yeast growth (white streaks) or fermentation (bubbles) were observed among the vastly diverse microbial population on the media. After incubation, the samples were vortexed to mix and 100 μL were transferred and spread onto enrichment plates containing 20 g agar, 20 g sucrose, and 6.7 g yeast nitrogen base per 1 L. Plates were incubated at 30 °C for 3–5 days, or until yeast-like colonies had formed. Yeast from the spontaneous fermentation were directly plated onto enrichment plates.

### 2.3. Identification of Isolates

Isolates were chosen for identification based on color, shape, texture, and opacity. Yeast colonies were typically white-yellow, round, smooth or rough, and opaque. Colonies fitting this description were streaked for single colonies onto rich media plates. Typically, we isolated one colony per sample. However, in rare cases, if colonies exhibited different morphologies, we sequenced both. If the two colonies were later identified as the same species and had the same relative growth rate in 8 different conditions (11 chemical/media combinations at two different concentrations of chemical), then only one was retained in the analysis. Genomic DNA was extracted and then PCR amplified for the internal transcribed spacer (ITS) or D1/D2 regions of the rDNA. Both ITS and D1/D2 are regions traditionally used for sequencing of yeasts [49]. Genomic DNA was extracted using HIRTs lysis buffer [50] with the following modifications; 100 μL of zirconium beads were added to the cell pellet and buffer, and vortexed for 3 min to lyse cells before being centrifuged. PCR amplification of the ITS was performed using primers that encompassed the entirety of the ITS region. The 5′ primer sequence was (5′-TCCGTAGGTGAACCTGCGG-3′), the 3′ primer sequence was (5′-TCCTCCGCTTATTGATATGC-3′) and the typical PCR product was 841 bp long [51]. PCR reactions were set up at a volume of 50 μL, containing 1x Hi-Fi reaction buffer (included with enzyme), 5 μM of each primer, 2 mM mgCl$_2$, 1 mM dNTP cocktail, 3 μL DMSO, 1μL velocity DNA polymerase, and 50 ng of the purified gDNA from

the strain in question. PCR thermocycler conditions were as follows: 98 °C for 2 min, 25 × [98 °C for 30 s, 55 °C for 30 s, 72 °C for 40 s], 72 °C for 7 min. PCR amplification of the D1/D2 region was performed using primers and protocols [52]. The NL1 (5′-GCATATCAATAAGCGGAGGAAAAG-3′) and NL4 (5′-GGTCCGTGTTTCAAGACGG-3′) primers were used. The reactions were performed in 25 μL aliquots containing 20 mM Tris-HCl (pH 8.4), 50 mM KCl, 2.5 mM mg2 Cl, 5% DMSO, 0.5 mM dNTPs, 0.1 μM each of the primers, 1 μLTaq DNA polymerase, and 50 ng of purified gDNA. PCR cycler conditions were as follows: 95 °C for 7 min, followed by 40 × (95 °C for 60 s), 53 °C for 2 min, 72 °C for 1 min, with a final extension at 72 °C for 10 min. PCR products were run out on a 1% agarose gel and extracted using a Bioline Isolate II PCR and Gel Kit. Sanger Sequencing of the amplified ITS and D1/D2 DNA was performed in the West Virginia University Genomics Core Facility. Sequences were viewed in Geospiza's FinchTV and searched in NCBI BLAST database, where e-value, query coverage, and identity percentage were used to decide the identity of the sequenced colony.

## 2.4. Phylogenetic Analysis

18S rRNA and 26S rRNA sequences were downloaded from NCBI nucleotide database for one representative species per genus. 18S rRNA and 26S rRNA sequences corresponding to the same genus were concatenated as single sequences. The concatenated sequences were aligned with muscle [53]. The alignment was used as input in the generation of a maximum likelihood tree via MEGA [54]. Initial tree(s) for the heuristic search were obtained automatically by applying neighbor-joining and BioNJ algorithms to a matrix of pairwise distances estimated using the maximum composite likelihood (MCL) approach, and then selecting the topology with superior log likelihood value. The reliability of a phylogenetic tree was assessed with a bootstrap test (500 iterations). The length of a branch denotes the genetic distance (e.g., number of substitutions per unit time) between the two taxa it connects. Maximum likelihood trees were created by location when at least three different genera were isolated thereby following the same procedure explained above. Normalized phylogenetic trees were calculated as the ratio between the total branch length and the number of different genera per location [8]. For locations with less than three genera, the normalized phylogenetic diversity is zero.

## 2.5. Phenotypic Characterization

Yeast growth on serial dilution media with glyphosate-based herbicide (GBH) were scored relative to growth on rich and minimal media alone [55]. Briefly, yeast were grown overnight to saturation in YPD. The equal amounts of yeast were diluted ten-fold four times onto appropriate plates. RM11, a GBH, and copper resistant strain; and YJM789, a GBH, and copper sensitive strain, were spotted on to each plate as controls. Each strain was scored 0–4 depending on how many spots grew. Credit41 is the glyphosate-based herbicide used here and diluted according to the final percentage of glyphosate in the media. GBH was implemented into the three types of media; YPD, YM, and WYF. Yeast extract peptone dextrose (YPD) contains 20 g yeast extract, 20 g peptone, 2% dextrose, and 20 g agar per liter. Yeast minimal media contains (YM) 20 g yeast extract, 1.7 g nitrogen base without amino acids or ammonium sulfate (BD catalog number 233520), 1 g of ammonium sulfate, and 20 g agar per liter. WYF is YM with tryptophan, tyrosine and phenylanine supplemented. Copper sulfate was implemented in YM selective media at concentrations of 200–800 μM. Growth of wild yeasts were scored on the second day of growth across all conditions with YJM789 and RM11 serving as controls with low growth and high growth, respectively. Locations with fewer than three *S. cerevisiae* were filtered from analysis. *S. cerevisiae* were clustered with the hclust function in package stat, MetaboAnalyst normalized, clustered with the hclust function in the package to generate the heatmaps. Barplots (done with R package ggpubr 0.2.5), showing the mean scores values and the standard errors, per location (or source) and condition. We also calculated *p*-values using a student T tests which are presented in the supplemental tables.

## 3. Results/Discussion

### 3.1. Genetic Variation of Fungicide in Geographically Diverse S. cerevisiae

Six strains had been shown to vary in response to glyphosate in the Credit41 formulation [26]. These strains represented the major sources of *S. cerevisiae* and included isolates with backgrounds in agriculture, forests, clinics, and laboratories. The two agricultural isolates grew well in the presence of glyphosate in all conditions tested. To determine if this was a general characteristic of agriculture associated yeast, we expanded the number of strains. The yeast from the historical collection dated back over 100 years. A total of 35 strains with known background information including year and source of isolation from agricultural, forest, or clinical sources were tested on YPD, YM, and WYF with glyphosate (Figure 1A and Supplemental Table S1). Most of the strains tested here were agricultural isolates from after the commercial release of Roundup in 1974. In YPD, the shikimate pathway is downregulated and so the growth inhibition is primarily from the effects of the additives which affect the cell wall [27]. The relative resistance to glyphosate on YPD was evenly distributed across yeast from different environments before and after 1974. No strain isolated before 1974 could grow well in the presence of 0.25% glyphosate on YM when yeast was dependent on the shikimate pathway to produce all the aromatic amino acids, para-aminobenzoic acid, and Coenzyme Q10. However, 8 out of the 14 strains tested that were isolated after 1974 had high resistance (defined as no change in growth at 0.25% glyphosate YM compared to YM). The resistance yeasts collected before 1974 were statically different from those collected after (*p*-value = 0.0004 via student T test). The addition of WYF supplements bypasses the glyphosate inhibition of the shikimate pathway by providing the essential aromatic amino acids. However, growth inhibition in WYF media also suggests growth inhibition from the additives or from inhibition of a non-canonical glyphosate target. There was still as correlation between year of isolation and growth in the presence of WYF and GBH (*p*-value = 0.007). Forest and clinical isolates did not show the same pattern and were sensitive or only moderately resistant to glyphosate.

Glyphosate is degraded by bacteria into non-active compounds, and it is short-lived in the soil [56]. However, metal-based fungicides accumulate in the soil altering microbial communities [57]. With widespread use, glyphosate and its breakdown products have been detected in numerous water sources, soils, and precipitation [58]. The exposure of agricultural yeast to glyphosate has been assumed given the popularity of the herbicide. Copper, the active ingredient in the Bordeaux mixture, along with slaked lime inhibits fungal growth by the production of reactive oxygen species which represses the germination of fungal spores. Yeast express the metallothionein, Cup1, that binds and sequesters copper. Copper resistant yeast often amplify the *CUP1* locus [59] and alter zinc levels [38]. However, in contrast to GBH, there was only sporadic resistance to copper in the historical collection and there was no temporal bias as seen in the agricultural strains with glyphosate collected before and after 1974 (*p*-value = 0.37) (Figure 1B). Additionally, some copper resistance was found in the clinical isolates, but none was noted in forest isolates, which suggests higher copy numbers of *CUP1* in clinical isolates.

### 3.2. Isolate Identification

Yeast from the historical collection often only presented one strain isolated from a single location, but strains were collected from all over the world and the previous herbicide exposure was difficult to determine. We strongly suspected that agricultural yeast isolated in the 1980s and later had been exposed to glyphosate. We sought to isolate yeast from areas with varying amounts of glyphosate exposure. There is an established relationship between *Saccharomyces cerevisiae* and oak trees, but for the sampling purposes of this study both oak and non-oak based substrates were collected in order to encourage collection and isolation of more genetically varied yeast. Sampling took place at 15 different locations throughout the study and most heavily surveyed these locations between the months of May and September, as we found this season had the highest rates of isolation success (Figure 2 and Supplemental Table S2). We also chose different types of locations. Chestnut Brew Works was located

in a suburban neighborhood and spent mash from the beer brewing process was left outside where deer and other herbivores were known to feed on it. WVU campuses landscaping was managed by conventional means. The WVU Core Arboretum is a 91-acre temperate deciduous forest on the banks of the Monongahela River and was established in 1948. The area is dominated by native trees and plants. The WVU Organic Farm was converted from a conventional farm in 1989 and is located within the city limits of Morgantown, WV. Coopers Rock overlooks the Cheat River 13 miles outside of Morgantown and has been a state park for over 80 years. Osage mine is a remediated strip coal mine that is now mainly an open field with a second-growth forest surrounding it. Outside the boundary of the mine is a conventional farm with cows and is separated by a barbed-wire fence. Bear and deer scat were also present and sampled. In addition to Granville, WV, all these areas are in Monongalia County. The Monongahela National Forest was established in 1920 and is the home to the Fernow Experimental Forest, Spruce Knob, and the Sinks of Gandy. To the South, the Appalachian Trail runs through the George Washington and Jefferson National Forest. A home garden in Chapala in the state of Jalisco in Mexico represented a typical suburban home.

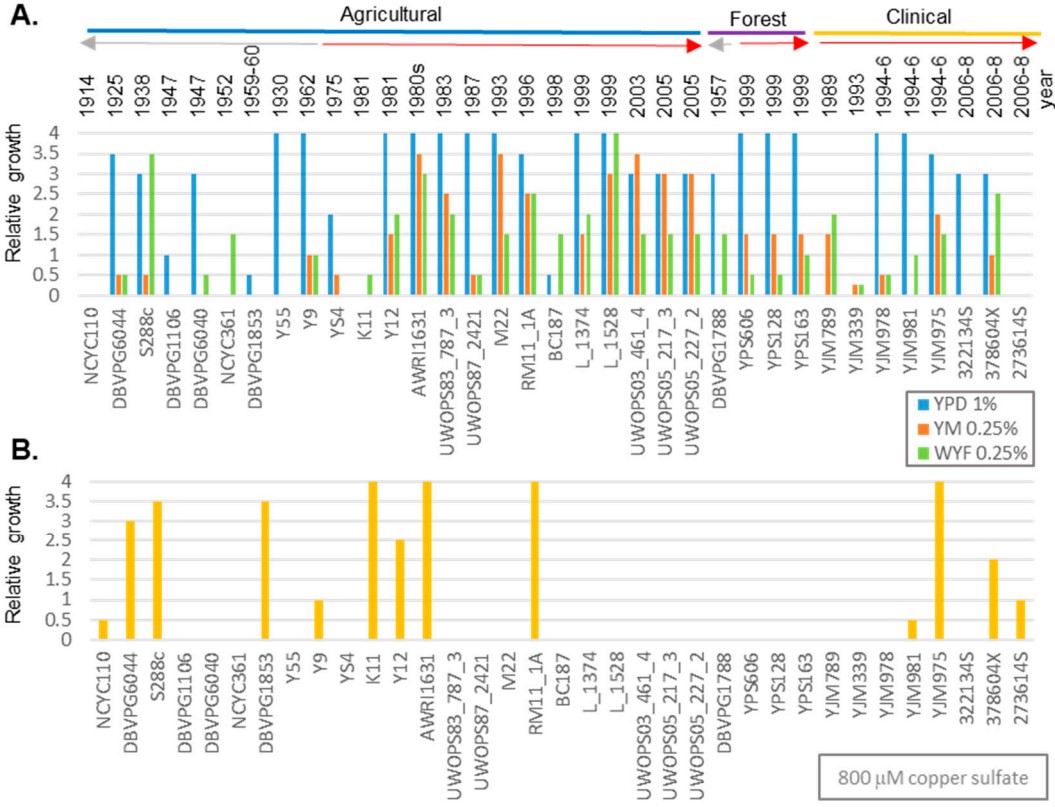

**Figure 1.** Fungicide response by the historical yeast collection. Relative growth of each strain was compared to RM11 and YJM789. Growth similar to untreated yeast on each media was a rated as four and no detectable growth was rated as zero. (**A**) Quantification of serial dilution of yeast grown on YPD (rich media) with 1% glyphosate (blue), YM (minimal media) with 0.25% glyphosate (orange), and WYF (yeast minimal media supplemented with aromatic amino acids) with 0.25% glyphosate (green). Credit41, the commercial glyphosate-based herbicide was used as the source of glyphosate. No growth was scored as zero and full growth was scored as four. (**B**) Quantification of serial dilution of yeast grown copper 800 μM copper sulfate in YM.

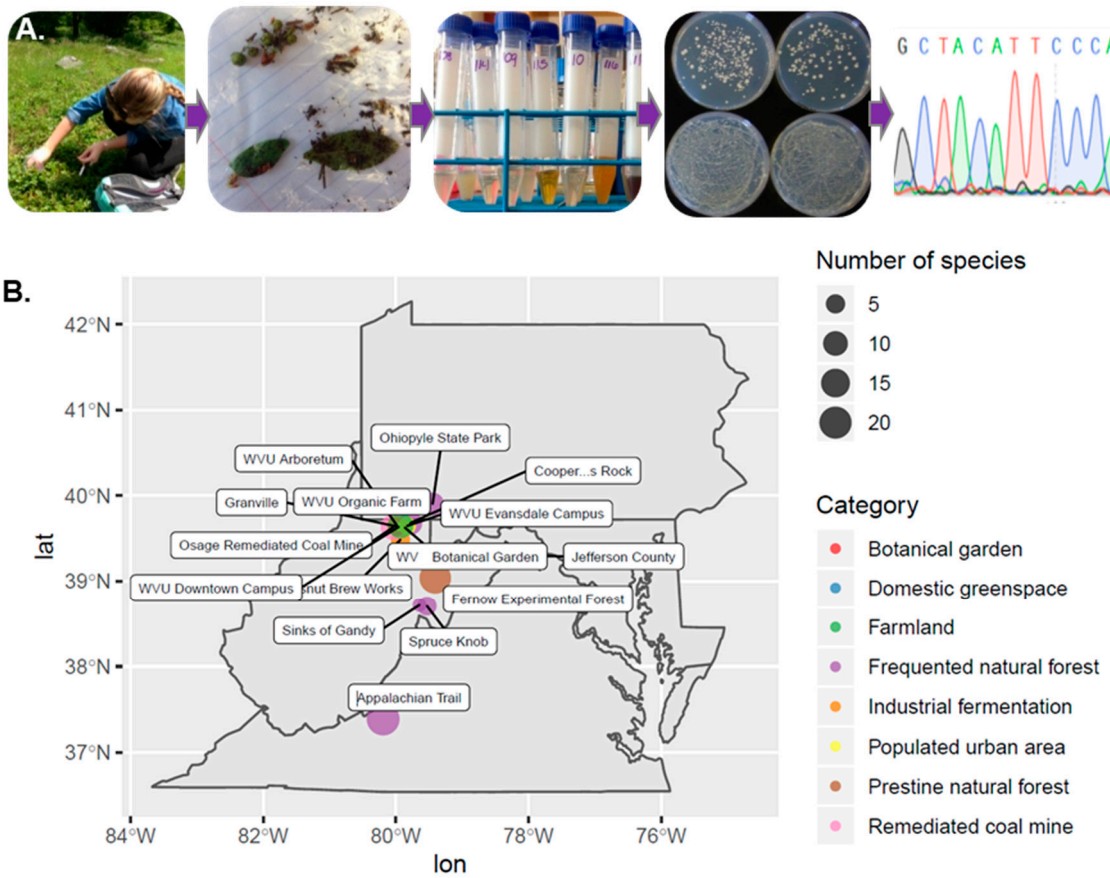

**Figure 2.** Geographic diversity of sampling sites across Appalachia. (**A**) Workflow of isolated wild yeast from environmental samples. Samples were collected in the field in plastic bags, enrichment media solubilized the sample, and then the liquid and particulates were transferred to sterile 15 ml falcon tubes for 1–2 weeks. The liquid was then plated onto solid media to isolate single colonies. The genomic DNA was extracted, and the rDNA region was sequenced. (**B**) Environmental samples were collected in these locations. The size of the circles notes how many unique species were identified from each location. The color of the circle notes the type of location.

A total 561 collections and enrichments were completed. Most of the incubations yielded yeast-like colonies. In some cases, a single incubated sample showed more than one type of yeast-like colony on media, such as a wrinkled or smooth phenotype. Antibiotics, ethanol, sucrose, and malt were added to the enrichment media to discourage bacteria and mold growth. A combination of D1/D2 primers to the 26S region and primers specific to the ITS region were used to identify species. The ITS region is one of the most commonly used regions of the fungal genome for species identification [60] because of several factors. In particular, the ITS region is present in all organisms, it repeats within the organism, and it is a region that contains highly conserved and highly variable sites. These sites within the ITS region allows primers to target the conserved regions while analysis of the variable regions determined the species of fungus [60]. This area can be used to determine species precisely, but the GenBank database is lacking in sequence entries for some fungal species [61]. Sequences that had less than 99% identify were dropped from the study. Sequencing of the ITS region has more resolution but the published library of D1/D2 sequences is more complete. This region is less genetically variable than the ITS region, especially for closely related species, but it is more completely filled with regards to the larger population [61]. A total of 237 yeast were identified, representing 23 genera (Figure 3A). Of the yeast isolated and identified, the *Saccharomyces* genus was isolated the most often, followed by *Metschnikowia*, *Kluyveromyces*, *Pichia*, *Lachancea*, and *Hanseniaspora* (Figure 3A). Species identified included *Kluyveromyces lactis*, *Lachancea fermentati*, *Pichia kudriavzevii*, *Pichia manshurica*,

*Pichia membranifaciens, Saccharomyces cerevisiae, Saccharomyces paradoxus*, and *Wickerhamomyces anomalus* (Supplemental Table S2). In cases were species identification was ambiguous, only the genus was noted. We expected *S. cerevisiae* and *S. paradoxus* to be isolated together because of their association on oak trees and similar niche environments. We found that multiple species could inhabit the same environments simultaneously and survive the competition of the enrichment process designed to enrich fermenting yeast.

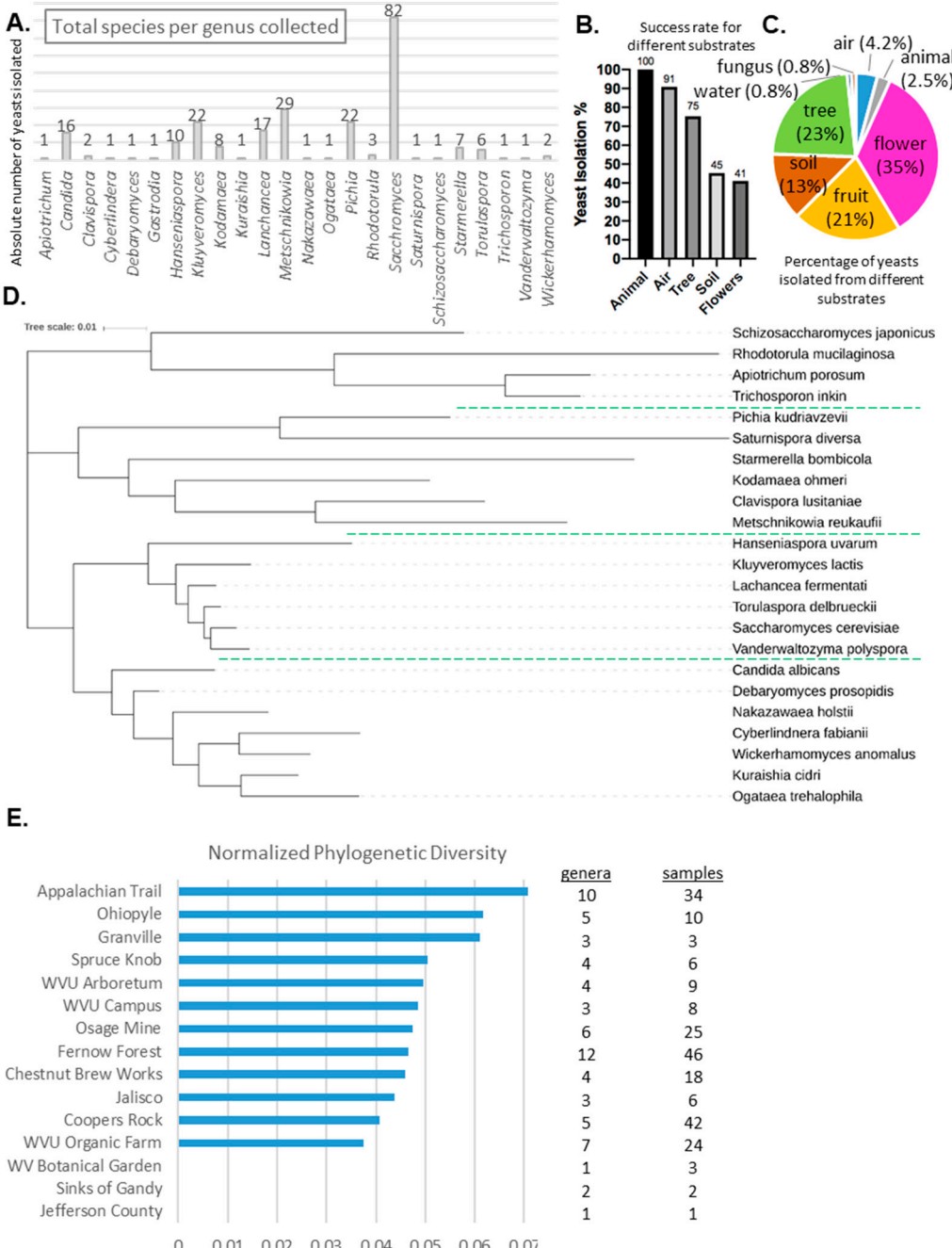

**Figure 3.** Phylogenetic diversity of yeast isolated from environmental samples. (**A**) The number species identified from each genus were graphed. (**B**) Success rate of isolating different species from different substrates. (**C**) Comparison of different types of substrates that that contained isolated yeast species. (**D**) Phylogenetic tree of genera identified. One representative species from each genus was used. (**E**) Phylogenetic diversity of each location normalized by the number of individual species identified. The number of genera identified and the number of unique yeast species from the samples are noted.

In terms of yeast isolation success, animal and air substrates produced the highest percentage of successfully isolated yeasts, with 100% and 91% isolation success respectively (Figure 3B). However, a low quantity of these types of substrates were collected to begin with, as only four animal substrates and 11 air substrates were processed. An individual sample was also capable of producing multiple yeast-like colonies. Therefore, both these percentages are inflated from not only their small sample pool, but also due to a signal animal substrate being able to produce three different yeast species. Not all samples collected had their substrates recorded (Summer Outreach program students at the WVU Organic Farm and Osage Mine); however, both of these locations were sampled at other times. Trees on the other hand had both a high number of sample substrate collected of 93 and produced an isolated species of yeast in three of every four samples (75%). Flower substrates had the highest amount of substrate collected as they were the easiest to do so by being light and simple to extract from the locations surveyed. Surprisingly, flowers yielded the lowest percentage of yeast species isolated at 41%. By absolute numbers flowers, trees, fruit, and soil yielded the most yeast (Figure 3C). Insects are postulated to be yeast vectors given the volatiles of fermentation attract insects and provide essential B vitamins to the developing larvae. In *Drosophila* isolated from California wineries, fungal communities are composed of *Hanseniaspora uvarum* (30%), *Pichia manshurica* (12%), and *S. cerevisiae* (9%) [62]. From our brewery isolations, wort and boiled barley left outside for a week grew *Pichia kudriavzevii* and *S. cerevisiae* 5 and 10 times, respectively. We also found *Pichia membranifaciens* and *Wickerhamomyces anomalus* were once on each substrate. *Kluyveromyces lactis* was found in a nearby tree at the brewery. A phylogenetic tree of all the genera identified demonstrates the genetic diversity of the yeast isolated (Figure 3D). For each genus, the species isolated the most was selected to represent that genus. Although enrichment was designed specifically for *Saccharomyces*, genetically diverse yeast species were isolated. This supports diverse yeasts' ability for survival in high alcohol environments with fermentable carbon sources. Phylogenetic diversity was calculated by adding the branch length and dividing by the number of genera identified at each location. The conventional farm in Jefferson county had no diversity as all the yeast identified were *S. cerevisiae* (Figure 3E). Areas that were deemed pristine, including the Fernow Forest and Appalachian Trail, had the most phylogenetic diversity. Phylogenetic diversity was impacted by the number of samples and how heavily the area was sampled.

To determine if known glyphosate resistance of *S. cerevisiae* correlated with the location the growth of yeast in two different concentrations of glyphosate-based herbicides was measured. *S. cerevisiae* was found at 11 of the sites and the site with the most found at Coopers Rock, followed by Osage Mine, and then Chestnut Brew Works (Figure 4A). Consistent with previous studies, trees were the preferred environment for *S. cerevisiae*. Approximately half of the *S. cerevisiae* isolates were harvested from flowers, fruit, and soil sources (Figure 4B). At Chestnut Brew Works, the spontaneous fermentations (air) always yielded *S. cerevisiae* with *Pichia kudriavzevii* or *membranifaciens* (Supplemental Table S2).

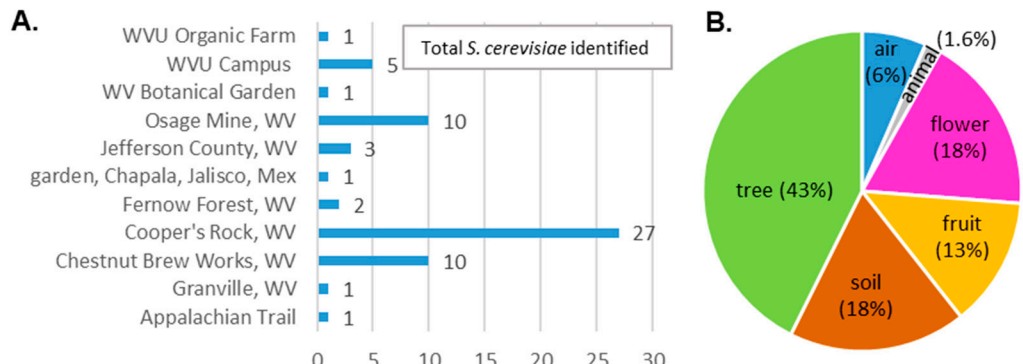

**Figure 4.** Characterization of *S. cerevisiae* isolation. (**A**) Distribution of *S. cerevisiae* isolated from different locations. (**B**) Distribution *S. cerevisiae* isolated from different sources.

The original purpose for collecting wild *S. cerevisiae* was to determine if routine GBH exposure would lead to increased resistance. From the historical collection we noted that *S. cerevisiae* isolated from agricultural sources in the 1980s and later tended to be more resistant to GBH but there was no trend noticed for copper resistance. We used Credit41 rather than pure glyphosate because that was the glyphosate-based herbicide used at mine (Figure 5A and Supplemental Table S3). Five locations with more than three *S. cerevisiae* isolates were subject to hierarchical clustering. One of the most surprising findings during the characterization of the isolates was that a third of strains isolated from Coopers Rock State park displayed high resistance to glyphosate, while the majority of *S. cerevisiae* collected from the state park showed little to no resistance. According to the West Virginia Division of Forestry, there have been several incidences of herbicide use in Coopers Rock. The first application was during a grapevine herbicide study conducted by the USDA-Forest Service, Timber, and Watershed Laboratory in 1978 that ran until 1980 [63]. Additionally, there is also a power line that runs through the Coopers Rock area and crosses the Cheat River, which has been the target of aerial herbicide treatments to protect the integrity of the power line structure. Yeast from Chestnut Brew Works did not cluster, while yeast from Jefferson County farm had nearly identical phenotypic profiles and were generally resistant to glyphosate. The average resistance was highest at the Jefferson County site because no glyphosate sensitive *S. cerevisiae* were identified. In comparison to growth at Coopers Rock, yeast isolated from Jefferson County were most resistant to glyphosate in YPD (*p*-value = 0.15), followed by WYF and YM but these comparisons were not statistically significant (*p*-value = 0.16 and 0.22, respectively). The relative growth compared to RM11 and YM789 facilitated the visualization the phenotypic diversity (Figure 5B). Following Jefferson County, yeast from WVU campus had less variation in phenotype profile than other locations. PCA according to sample type did not partition yeast with similar phenotypic profiles (Figure 5C). Because we were only sequencing the rDNA region, we could not rule out that we had isolated genetically identical yeast. We measured the copper sulfate resistance of the same yeast, as genetically identical yeast would also have the same copper sulfate resistance. There was no correlation between *S. cerevisiae* glyphosate resistance and copper resistance. None of the areas were known to have copper sulfate applied and only four yeast could tolerate high levels of copper sulfate (Figure 5A). Half of the copper resistant yeast were isolated from the WVU campus and the rest were isolated at Coopers Rock.

Previous glyphosate exposure was based on personal communications with state park employees and farm owners. The owners of the Osage mine permitted an experimental poplar tree plantation on the site [64] and they began controlling the growth of weeds in 2013 by application of numerous herbicides, including Credit41. Sampling at the mine began in 2014. The tree plantation was in the middle of an open field surrounded by a second-growth forest (Figure 6A). Yeast were sampled from within a field dominated by raspberry bushes and the surrounding trees yielded the most *S. cerevisiae* (Supplemental Table S2). Hierarchical clustering and PCA could partially separate yeast isolated from the field and forest (Figure 6B). From the heatmap analysis suggested that the yeast isolated from the trees in the surrounding forest were more phenotypically diverse than the field. The smaller clustering of the field yeast is evidence of yeast that have migrated from trees, potentially from insect vectors. Field yeast had less resistance to glyphosate and the *p*-values for the highest glyphosate concentration for YPD, YM, WYF respectively were 0.013, 0.067, and 0.016. As the application of glyphosate was recent and the growth of plants in the field was recent, the lower phenotypic diversity could represent several possibilities. There could be a selection process like the conventional farm in Jefferson County, where a stressful environment reduced the genetic diversity in the field so that the existing yeast that happen to tolerate glyphosate were only present in the forest. There were other herbicides oxyfluorfen, pendimethalin, and clopyralid applied to the field. Resistance to those herbicides may have been a more potent selection than glyphosate. We did not detect genetic variation in resistance to these herbicides the historical collection and were not further tested in wild isolates.

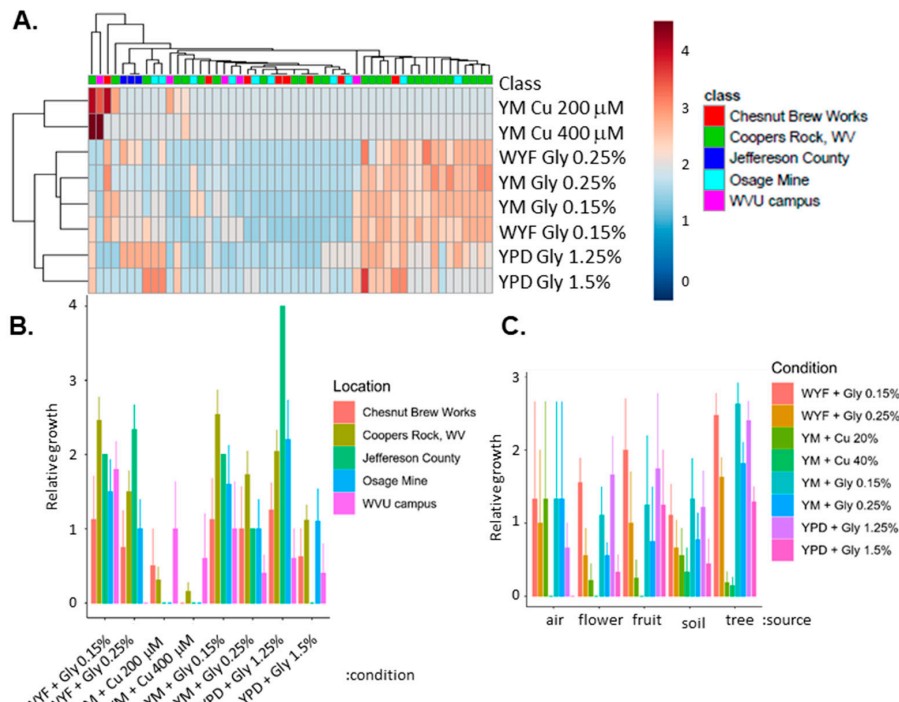

**Figure 5.** Phenotypic differences of *S. cerevisiae* to different herbicide treatments. (**A**) Heatmap of hierarchical clustering of *S. cerevisiae* grown in eight different conditions. Yeast were serial diluted and grown on rich media (YPD), minimal media (YM), and minimal media supplemented with aromatic amino acids (WYF) with the noted amount of glyphosate w/v from the glyphosate-based herbicide (Credit41) or 200–400 μM copper sulfate. Red noted robust growth while blue noted less growth. (**B**) Bar graph of *S. cerevisiae* growth in different herbicides across different locations. (**C**) Bar graph of *S. cerevisiae* growth in different herbicides across different sample sources. The explained variance is noted in parentheses.

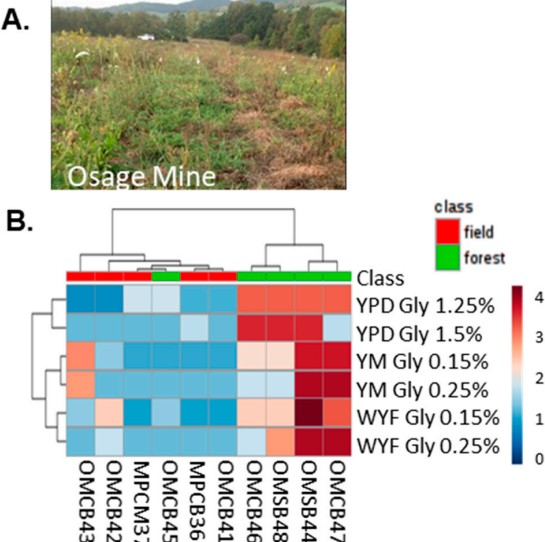

**Figure 6.** Phenotypic variation in *S. cerevisiae* from Osage Mine to glyphosate. (**A**) Osage mine during the summer after spraying with glyphosate (Credit41). (**B**) Heatmap of hierarchical clustering of *S. cerevisiae* grown in glyphosate. Yeast were serial diluted and grown on rich media (YPD), minimal media (YM), and minimal media supplemented with aromatic amino acids (WYF) with the noted amount of glyphosate w/v from the commercial formulation (Credit41).

## 4. Conclusions

The enrichment process isolated a genetically diverse group of yeast. We had expected that the selective isolation protocol would isolate more *S. cerevisiae* because of the moderately high levels of alcohol, malt, and the 25 °C enrichment periods, which would allow *S. cerevisiae* to out-compete other species [15]. However, we found a large diversity of the types of yeast isolated with this method. We also found that on a large scale, trees are a substrate source rich in yeast populations and consistent with prior studies that they are a preferred environment for *S. cerevisiae.* Of the 237 identified isolates, only 26% (62) were identified as *S. cerevisiae*. Of all the locations, the majority (27) of *S. cerevisiae* isolates came from Coopers Rock State Park in West Virginia. Of the 23 different genera, pristine areas—such as the Appalachian Trail and the Fernow Forest—were the most phylogenetic diversity across the 15 locations surveyed. *Saccharomyces, Pichia, Kluyveromyces, Candida, Metschinikowia*, and *Lachancea* were the primary genera found. Similar to previously identified agricultural isolates, yeast isolates from RoundUp Ready™ corn on a conventional farm in Jefferson County, WV had the highest average of glyphosate resistance. Samples collected from Cooper's Rock, WV had the highest resistance in unique samples, and contained the most resistance diversity of *S. cerevisiae* collected from each location. Between genetically diverse yeast, there are tens of thousands SNPs [29], quantitative trait loci analysis of glyphosate resistance shows that it is a polygenetic trait [26] and there are multiple changes that can occur to select for glyphosate resistance [27].

**Supplementary Materials:** The following are available online at http://www.mdpi.com/1424-2818/12/4/139/s1, Table S1: Glyphosate and copper resistance for historical collection of S. cerevisiae. Growth was scored using RM11 and YJM789 as examples of resistant and sensitive strains on every plate. Yeast were grown for 2 days and scored for growth on 0.15% and 0.25% of glyphosate from the Credit41 formulation in YM and WYF and 1.25% and 1.5% glyphosate in YPD. Copper resistance was tested at 800 μM of copper sulfate. Table S2: Wild yeast collection. For each isolate sample date, location, coordinates, collected by, source material, genus, species, identification by ascension number, associated E-value, query coverage, ID %, and whether ITS/D1D2 was used are listed. Table S3: Glyphosate and copper resistance for isolated S. cerevisiae. Growth was scored using RM11 and YJM789 as examples of resistant and sensitive strains on every plate. Yeast were grown for 2 days and scored for growth on 0.15% and 0.25% of glyphosate from the Credit41 formulation in YM and WYF and 1.25% and 1.5% glyphosate in YPD. Copper resistance was tested at 200 μM and 400 μM of copper sulfate.

**Author Contributions:** Conceptualization, J.E.G.G.; Methodology, C.B.B.; Validation, J.B.B.; Formal Analysis, J.B.B. and A.P.; Investigation, J.B.B., M.J.W., and C.B.B.; Data Curation, J.B.B.; Writing—original draft preparation, J.B.B.; Writing—review and editing, J.E.G.G.; Visualization, A.P. and J.E.G.G.; Supervision, J.E.G.G.; Project Administration, J.E.G.G.; Funding Acquisition, J.E.G.G. All authors have read and agreed to the published version of the manuscript.

**Funding:** National Science Foundation MCB 1614573, DEB 0417678, DEB 1019522.

**Acknowledgments:** We would like to acknowledge the Department of Biology, WVU Honors College, and WVU Genomics Core Facility, Morgantown WV for the support provided to help make this publication possible. We thank Luke Evans who suggested the Osage Mine as a location and applied the glyphosate-based herbicide to the field. Daniel Panaccione and Steve DiFazio provided insights into fungal and genetic diversity. The high school students from the NSF TRiO Upward bound and the general public who attend I ASK WHY (Information Acquired by Students who Know West Virginia Has Yeast), Taizina Momtareen, J. Philip Creamer, Michael Ayers, Apoorva Ravishankar, Matthew Pyster, Audrey Biega, Mahmoud Summers, and Samatha Jusino who assisted in the collection of wild isolates. Kirsten McNeal tested the historical yeast. West Virginia University PSCoR and West Virginia University Senate Grant provided initial funding. This work was funded by the National Science Foundation (grant No. MCB 1614573) to JEGG. The Fernow Forest is maintained by the National Science Foundation from their Long-Term Research in Environmental Biology program (grant Nos. DEB-0417678 and DEB-1019522). The funding agencies had no role in the study design, the collection, analysis and interpretation of data; the writing of the report; or the decision to submit the paper for publication.

**Conflicts of Interest:** The authors declare no conflict of interest. The sponsors had no role in the design, execution, interpretation, or writing of the study.

**Data Availability:** Strains and plasmids are available upon request. The authors affirm that all data necessary for confirming the conclusions of the article are present within the article, figures, and tables.

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
