# Peer review of "The Yeast Atlas of Appalachia: Species and Phenotypic Diversity of Herbicide Resistance in Wild Yeast"

_diversity, doi:10.3390/d12040139_

Round 1

Reviewer 1 Report

This is an interesting manuscript that is partly based on a ‘citizen science’ project to isolate wild yeasts from soils and plants in the Appalachia region of the USA. It consists of two rather separate parts. First, the authors analyzed resistance/sensitivity to the herbicide glyphosate (Roundup) in Saccharomyces cerevisiae isolates in a well-studied ‘historical collection’ of isolates (from Liti et al 2009, ref. 2), which shows that S. cerevisiae strains that were isolated from agricultural sites more recently than the commercialization of Roundup in 1974 tend to be more resistant to Roundup. Second, the authors isolated more than 200 strains of wild yeasts from very diverse locations in Appalachia (some pristine, some agricultural, some polluted, etc, including sites with a known history of glyphosate treatment). The primary goal of this second part was (I think) to isolate S. cerevisiae strains from these sites to see if there were correlations between their source and their levels of glyphosate resistance. However, relatively few of the Appalachia yeasts isolated turned out to be S. cerevisiae, which limited the extent of analysis that could be done. Nevertheless, despite its disjoint nature, the study provides a useful and detailed survey of the fermentative yeasts that are present in Appalachia. There are relatively few surveys or atlases of this type; the best-known other ones are by Sylvester et al (ref. 15) and Groenewald et al (PMID 30016423).

My main critical comment is that the two parts of the study are not very well integrated. It is difficult to compare the analysis of glyphosate resistance in the Appalachia S. cerevisiae (Fig. 5A) to that in the historical collection (Fig. 1A) because different methods seem to have been used in the two figures. Does Fig 4A include some control strains from the set in Fig. 1A, and if so where are they? I don’t understand why the scale in Fig 1A runs from 0-4, but in Fig 5A from -4 to +4. 

L408: The statement in the Conclusions that “yeast isolates from a conventional farm in Jefferson County WV had the highest glyphosate resistance” seems to come completely out of left field. Unless I am misreading Fig 5A, this figure shows LOW glyphosate resistance (YM Glyph) in the Jefferson County samples, compared for example to the Coopers Rock samples. And the corresponding Results section (L343-359) highlights the high glyphosate resistance in the Coopers Rock samples and doesn’t mention Jefferson County.

L320-322. I don’t understand the description of how the phylogenetic diversity of each location was calculated (to get Fig. 3D). Surely dividing by the number of genera present at a location means that locations with more genera will have a lower diversity index, which is completely counter-intuitive. Is there a reference for this method?

L124 mentions an interesting conclusion that flowers yielded the most yeast species diversity. I think that this is supposed to summarize the results in Fig 3B and L309. However, the pie chart in Fig 3B just shows the number of sample types that contained a yeast, so if you sampled lots of flowers it is not surprising that the flower part of the pie is the biggest. It would be much more informative to show the ‘yeast success rate’ for each type of substrate, i.e. what percentage of flower samples successfully yielded a yeast, versus other types of sample? That information could be a useful guide for people trying to find wild yeasts in the future. Without this sort of analysis, the conclusion that flowers yielded the most diversity is not warranted.

Other minor comments:

Are all the yeast strains isolated in this study (Table S1) available for distribution?

The words “yeasts” and “yeast” seem to be used interchangeably throughout the text, as plural nouns, which I found irritating. Please use “S. cerevisiae’’ where you specifically mean this species (e.g. in L18), and use “yeasts” when referring to samples that contain a variety of different species.

L14-18: The first 3-4 lines of the Abstract could be improved. The first sentence seems to be tacked on to a different abstract. S. cerevisiae is a singular noun.

L206: It would be useful to state the compositions of YPD (rich), YM (minimal) and WYF media here. I actually looked up ref. 48 to try to find a recipe for YM, but this reference does not contain information about media and doesn’t seem to be relevant to the sentence.

Typos:

L25, Kluyveromyces

L46, the genus (not the genera). And what does “diversity… has a complex history…” actually mean?

L156, Sniegowski et al should be [10]

L322, extra text at the end of the Figure legend

L371, poplar

Typos in Fig 3A: should be: Kuraishia, Nakazawaea

Author Response

R1.C1 My main critical comment is that the two parts of the study are not very well integrated. It is difficult to compare the analysis of glyphosate resistance in the Appalachia S. cerevisiae (Fig. 5A) to that in the historical collection (Fig. 1A) because different methods seem to have been used in the two figures. Does Fig 4A include some control strains from the set in Fig. 1A, and if so where are they? I don’t understand why the scale in Fig 1A runs from 0-4, but in Fig 5A from -4 to +4. 

R1.C2 L408: The statement in the Conclusions that “yeast isolates from a conventional farm in Jefferson County WV had the highest glyphosate resistance” seems to come completely out of left field. Unless I am misreading Fig 5A, this figure shows LOW glyphosate resistance (YM Glyph) in the Jefferson County samples, compared for example to the Coopers Rock samples. And the corresponding Results section (L343-359) highlights the high glyphosate resistance in the Coopers Rock samples and doesn’t mention Jefferson County.

R1.A2 We were averaging the resistance across all S. cerevisiae strains isolated from these sites. While Cooper’s Rock does have individuals with higher resistance than ones from Jefferson County, there were no sensitive strains isolated from Jefferson County. Cooper’s Rock had more diversity in glyphosate resistance which lowered overall the average resistance of S. cerevisiae isolated there.

R1.C3 L320-322. I don’t understand the description of how the phylogenetic diversity of each location was calculated (to get Fig. 3D). Surely dividing by the number of genera present at a location means that locations with more genera will have a lower diversity index, which is completely counter-intuitive. Is there a reference for this method?

R1.A3 The normalized phylogenetic diversity for each location was calculated as the ratio between the total branch length of the Maximum Likelihood tree for the location and number of total genera in the location. Locations with more genera does not necessarily has a lower index, as the index also depends of how different the genera are (estimated by the total branch length, that goes in the numerator). For example, as could be seen in Fig 3D, the Appalachian Trail, with 10 genera, has a larger normalized phylogenetic diversity than Ohiopyle, with 5 genera, which in turn has a higher diversity than Fernow Forrest with 12 genera. This is the reason we don’t use the total number of genera as the main diversity criteria [1].

R1.C4 L124 mentions an interesting conclusion that flowers yielded the most yeast species diversity. I think that this is supposed to summarize the results in Fig 3B and L309. However, the pie chart in Fig 3B just shows the number of sample types that contained a yeast, so if you sampled lots of flowers it is not surprising that the flower part of the pie is the biggest. It would be much more informative to show the ‘yeast success rate’ for each type of substrate, i.e. what percentage of flower samples successfully yielded a yeast, versus other types of sample? That information could be a useful guide for people trying to find wild yeasts in the future. Without this sort of analysis, the conclusion that flowers yielded the most diversity is not warranted.

R1.A4 Figure 3C (formally Figure 3B) was only for the number of S. cerevisiae isolated from different sources. We have added a new figure 3B is a graph of success rate for isolation of different yeast species from different substrates. We have also added text with improved explanation of the figure.

Other minor comments:

R1.A5 Are all the yeast strains isolated in this study (Table S1) available for distribution?

R1.C5 Yes, we have added text stated so.

R1.A6 The words “yeasts” and “yeast” seem to be used interchangeably throughout the text, as plural nouns, which I found irritating. Please use “S. cerevisiae’’ where you specifically mean this species (e.g. in L18), and use “yeasts” when referring to samples that contain a variety of different species.

C1.A6 We used yeast when only referring to S. cerevisiae and yeasts when referring to yeasts of different species. We have edited this to be more specific.

L14-18: The first 3-4 lines of the Abstract could be improved. The first sentence seems to be tacked on to a different abstract. S. cerevisiae is a singular noun.

revised

L206: It would be useful to state the compositions of YPD (rich), YM (minimal) and WYF media here. I actually looked up ref. 48 to try to find a recipe for YM, but this reference does not contain information about media and doesn’t seem to be relevant to the sentence.

The recipes for all the media have been added to the materials and methods.

Typos:

L25, Kluyveromyces - fixed

L46, the genus (not the genera). And what does “diversity… has a complex history…” actually mean? – genera is the plural of genus and has been edited to reflect that.

L156, Sniegowski et al should be [10] - fixed

L322, extra text at the end of the Figure legend - fixed

L371, poplar - fixed

Typos in Fig 3A: should be: Kuraishia, Nakazawaea - fixed

Reviewer 2 Report

The manuscript by Barney et al aims to assess the yeast diversity and glyphosate resistance in S. cerevisiae isolated from different environments which have been exposed to this compound, as well as sites where no records of glyphosate utilization were found. For this, the authors analyse an historical collection of S. cerevisiae strains and then isolate a large number of individuals from the field, and test for glyphosate resistance. The manuscript is well written and addresses an important question regarding yeast biodiversity. However, I still have major and minor comments that could help to improve the manuscript

Major comments:

  1. Throughout the text different comparisons are made between groups of yeasts, however no statistics were provided behind these analyses. In order to support many of the claims stated in the text, the authors should perform statistical analysis that validate the above mentioned differences. For example: diversity should not be an observation from a PCA analysis, but instead author could quantify the coefficient of variation for any tested group. This applies to several other comparisons.
  2. Many of the sections in the manuscript lack a proper description of what is being measured. For example, Figure 1 shows relative growth in the historical strains, yet no information is provided on how was this quantified and whether significant differences were found between groups.
  3. The introduction is vague and, although represents a nice bibliographic summary, there is no clear evidence of what the problem is. The first part seems an out-of-data state of the art, since many articles have been published in the past 20 years concerning genetic and phenotypic diversity in Saccharomyces. I suggest that the authors should better address the main subject of manuscript and update the text with more recent references.
  4. I am not sure whether the cupper section provides any significant insights to the manuscript. I suggest that the story should focus on glyphosate, this could help better addressing the main questions in the manuscript.
  5. Figures should contain self-explanatory captions. No information is provided in terms of what is being shown in the x-axis for most figures. Furthermore, no information is provided in the methods section on how these values were estimated.

Minor comments:

  1. Species names should be shown in italic
  2. How many colonies per tube were isolated and used?
  3. Media composition should be defined
  4. What does GBHs stands for?
  5. Adamo et al, 2012 is not in the correct format in the text
  6. Pie charts should also show percentage values.
  7. Figure 5A, what does the colour mean? How were these values obtained?

Author Response

1. Throughout the text different comparisons are made between groups of yeasts, however no statistics were provided behind these analyses. In order to support many of the claims stated in the text, the authors should perform statistical analysis that validate the above mentioned differences. For example: diversity should not be an observation from a PCA analysis, but instead author could quantify the coefficient of variation for any tested group. This applies to several other comparisons.

We used a metabolomics software which converts the relative growth rates to an arbitrary scale, we have converted the previous scale to align with the 0-4 scale presented in the supplemental tables. We do agree with the reviewer that the PCA representation for the growth scores is a poor choice and showing the coefficient of variation would not better either. We have repeated the analysis as barplots (done with R package ggpubr 0.2.5), showing the mean scores values and the standard errors, per location (or source) and condition. We also calculated p values using student T test.

2. Many of the sections in the manuscript lack a proper description of what is being measured. For example, Figure 1 shows relative growth in the historical strains, yet no information is provided on how was this quantified and whether significant differences were found between groups.

The relative growth scores are presented in Table S1. The yeast from both collections were grown in identical conditions with RM11 and YJM789 as controls on each plate to control for plate to plate variation and scoring between individuals. We used a metabolomics software which converts the relative growth rates to an arbitrary scale??. By definition, PCA plots do not have units.

3. The introduction is vague and, although represents a nice bibliographic summary, there is no clear evidence of what the problem is. The first part seems an out-of-data state of the art, since many articles have been published in the past 20 years concerning genetic and phenotypic diversity in Saccharomyces. I suggest that the authors should better address the main subject of manuscript and update the text with more recent references.

We added more references and clarified the purpose of the study

4. I am not sure whether the cupper section provides any significant insights to the manuscript. I suggest that the story should focus on glyphosate, this could help better addressing the main questions in the manuscript.

The purpose of the copper was two-fold. To ensure that areas with different GBH exposure had differences in growth that was not just better growth in any condition. We also used the copper resistance as well as the growth in GBH on various media as a way to determine if strains were duplicates from a single sample.

5. Figures should contain self-explanatory captions. No information is provided in terms of what is being shown in the x-axis for most figures. Furthermore, no information is provided in the methods section on how these values were estimated.

We improved the explanations and added y-axis labels were needed. For the x-axis in Figures 1A, 2A, 3A, 5A, and 6B are different yeast strains.

Minor comments:

  1. Species names should be shown in italic - done
  2. How many colonies per tube were isolated and used? – Typically, we isolated one colony per sample. However, in rare cases, if colonies exhibited different morphologies, we sequenced both. If the two colonies were later identified as the same species and had the same relative growth rate in eight different conditions (eleven chemicals/ media combinations at two different concentrations of chemical), then only one was retained in the analysis.
  3. Media composition should be defined - done
  4. What does GBHs stands for? - glyphosate-based herbicides
  5. Adamo et al, 2012 is not in the correct format in the text - fixed
  6. Pie charts should also show percentage values. – done
  7. Figure 5A, what does the colour mean? How were these values obtained? -Metabolanalyst generates the colors. In the heatmap, the redder strains grew more in the indicated condition while blue grew less. We have relabeled the scale, so it corresponds to the relative growth described in the materials and methods. The colored bars on the top refer to the location the yeast were isolated from.

Round 2

Reviewer 2 Report

The authors have answered all my comments. However, couple of them were not included in the text.

  1. Authors should incorporate how many colonies were chosen per tube (as mentioned in their answer) and the procedure followed at this stage.
  2. What metabolome software was used? This should also be included in the methods section and the protocol used for its implementation.

Author Response

We addressed the reviewer's comments.

  1. Authors should incorporate how many colonies were chosen per tube (as mentioned in their answer) and the procedure followed at this stage.

We inserted the description in the Methods section (lines 180-184).

  1. What metabolome software was used? This should also be included in the methods section and the protocol used for its implementation.

When we replaced the PCA plots with bar plots that information was also deleted. We have inserted the requested information (lines 236-238).